# Multi-Target Effects of Novel Synthetic Coumarin Derivatives Protecting Aβ-GFP SH-SY5Y Cells against Aβ Toxicity

**DOI:** 10.3390/cells10113095

**Published:** 2021-11-09

**Authors:** Ching-Chia Huang, Kuo-Hsuan Chang, Ya-Jen Chiu, Yi-Ru Chen, Tsai-Hui Lung, Hsiu Mei Hsieh-Li, Ming-Tsan Su, Ying-Chieh Sun, Chiung-Mei Chen, Wenwei Lin, Guey-Jen Lee-Chen

**Affiliations:** 1Department of Life Science, National Taiwan Normal University, Taipei 11677, Taiwan; gina851106@gmail.com (C.-C.H.); 781210@gapps.ntnu.edu.tw (Y.-J.C.); hmhsieh@ntnu.edu.tw (H.M.H.-L.); mtsu@ntnu.edu.tw (M.-T.S.); 2Department of Neurology, Chang Gung Memorial Hospital, School of Medicine, Chang Gung University, Taoyuan 33302, Taiwan; gophy5128@cgmh.org.tw; 3Department of Chemistry, National Taiwan Normal University, Taipei 11677, Taiwan; nobodyzmail@gmail.com (Y.-R.C.); baubauxyz@gmail.com (T.-H.L.); sun@ntnu.edu.tw (Y.-C.S.)

**Keywords:** Aβ, Alzheimer’s disease, coumarins, TRKB agonist, neuroprotection, therapeutics

## Abstract

Alzheimer’s disease (AD) is a common neurodegenerative disease presenting with progressive memory and cognitive impairments. One of the pathogenic mechanisms of AD is attributed to the aggregation of misfolded amyloid β (Aβ), which induces neurotoxicity by reducing the expression of brain-derived neurotrophic factor (BDNF) and its high-affinity receptor tropomyosin-related kinase B (TRKB) and increasing oxidative stress, caspase-1, and acetylcholinesterase (AChE) activities. Here, we have found the potential of two novel synthetic coumarin derivatives, ZN014 and ZN015, for the inhibition of Aβ and neuroprotection in SH-SY5Y neuroblastoma cell models for AD. In SH-SY5Y cells expressing the GFP-tagged Aβ-folding reporter, both ZN compounds reduced Aβ aggregation, oxidative stress, activities of caspase-1 and AChE, as well as increased neurite outgrowth. By activating TRKB-mediated extracellular signal-regulated kinase (ERK) and AKT serine/threonine kinase 1 (AKT) signaling, these two ZN compounds also upregulated the cAMP-response-element binding protein (CREB) and its downstream BDNF and anti-apoptotic B-cell lymphoma 2 (BCL2). Knockdown of TRKB attenuated the neuroprotective effects of ZN014 and ZN015. A parallel artificial membrane permeability assay showed that ZN014 and ZN015 could be characterized as blood–brain barrier permeable. Our results suggest ZN014 and ZN015 as novel therapeutic candidates for AD and demonstrate that ZN014 and ZN015 reduce Aβ neurotoxicity via pleiotropic mechanisms.

## 1. Introduction

Alzheimer’s disease (AD), the most prevalent type of neurodegenerative dementia, is characterized by progressive memory and cognitive impairments [1]. Extracellular accumulation of misfolded amyloid β (Aβ) in the brain (amyloid plaques) contributes to neuronal apoptosis, eventually leading to the shrinkage of the cortex and hippocampus. Aβ is produced from the cleavage of amyloid peptide precursor protein (APP) by β- and γ-secretases [2]. The Aβ tends to form oligomers and fibrils, which cause neuronal death by increasing oxidative stress, neuroinflammation, excitotoxicity, and apoptosis [3]. Among these mechanisms, Aβ-induced oxidative stress modifies proteins to perturb their biological function and impairs key biochemical and metabolic pathways in which these proteins normally play a role [4]. In addition, selective loss of acetylcholine-containing neurons in the brain contributes substantially to the cognitive decline in AD [5], and acetylcholinesterase (AChE) inhibitors modulating acetylcholine hydrolysis can increase the level and action duration of acetylcholine [6]. Accumulation of Aβ has also been proposed to be an activator to induce sequential pathological events such as the downregulation of the brain-derived neurotrophic factor (BDNF) signaling pathway [7,8].

BDNF, a member of the neurotrophic factor family, regulates the survival and differentiation of neurons by binding to its high-affinity receptor tropomyosin-related kinase B (TRKB) [9]. The binding of BDNF to TRKB induces the dimerization and autophosphorylation of TRKB [10] to activate the downstream extracellular signal-regulated kinase (ERK) and AKT serine/threonine kinase 1 (AKT). The phosphorylation of the cAMP-response-element binding protein (CREB) by ERK and AKT [11,12] further upregulates expressions of BDNF [13] and anti-apoptotic B-cell lymphoma 2 (BCL2) [14]. BCL2 binds to apoptosis regulator BCL2-associated X (BAX) to inhibit BAX-mediated mitochondrial outer membrane permeabilization, thereby inhibiting apoptosis [15,16]. The accumulation of oligomeric Aβ downregulates BDNF expression [17] and impairs the retrograde axonal transport of TRKB [18]. Intracerebral injection of BDNF in animal models of AD reduces Aβ-induced neurotoxicity and synaptic loss and improves memory impairments [19]. Therefore, the potentiation of the BDNF signaling pathway by TRKB agonists would be a strategy in treating AD.

Coumarins belong to a family of oxygen-containing heterocycles with a scaffold of 1,2-benzopyrone. These compounds exhibit diverse pharmacological effects such as reducing inflammation and oxidative stress and have been widely used as complementary and alternative medicines in treating neurodegenerative diseases [20]. It has been reported that derivatives of coumarin could prevent misfolded Aβ aggregation [21]. In AD cell and mouse models, synthetic coumarin–chalcone hybrid LM-031 demonstrates neuroprotective potential by regulating CREB and anti-oxidative pathways [22,23]. Coumarin derivative imperatorin also activates BDNF and CREB signaling to improve learning and memory deficits in prenatally stressed rats [24]. In addition, osthole lessens cognitive impairment in estrogen-deficiency mice by rescuing the reduction of BDNF and TRKB, as well as phosphorylation of CREB, in the hippocampus [25]. Here, we report the potential of two newly synthetic coumarins, ZN014 and ZN015, to reduce Aβ aggregations and oxidative stress as well as to enhance the TRKB signaling pathway in SH-SY5Y neuroblastoma cell models for AD.

## 2. Materials and Methods

### 2.1. Compounds and Cells

Coumarin derivatives 2-((3-benzoyl-2-oxo-2H-chromen-7-yl)oxy)propanoic acid (ZN009), 3-(4-chlorobenzoyl)-4-hydroxy-2*H*-chromen-2-one (ZN010) and 7-hydroxy-3-(4-methoxybenzoyl)- 2*H*-chromen-2-one (ZN011) were obtained from Enamine (Kyiv, Ukraine); ethyl 5-hydroxy-2-oxo- 2*H*-chromene-3-carboxylate (ZN014) and (*E*)-4-hydroxy-3-(3-(2-hydroxyphenyl)acryloyl)-2*H*- chromen-2-one (ZN015) were synthesized by modified procedures according to previously reported methodologies [22,26,27]. Curcumin and 7,8-DHF (positive controls) were purchased from Sigma-Aldrich (St. Louis, MO, USA). Human SH-SY5Y cells with inducible Aβ-GFP RNA expression [28] were cultured in Dulbecco’s modified Eagle’s medium (DMEM)-F12 supplemented with 10% fetal bovine serum (FBS) (Thermo Fisher Scientific, Waltham, MA, USA), 5 µg/mL blasticidin, and 100 µg/mL hygromycin (InvivoGen, San Diego, CA, USA).

### 2.2. Bioavailability and Blood–Brain Barrier (BBB) Permeation Prediction

Molecular weight, hydrogen bond donor/acceptor counts, octanol–water partition coefficient, and polar surface area of tested flavones were computed using ChemDraw (http://www.perkinelmer.com/tw/category/chemdraw, accessed on 30 October 2019). In addition, BBB permeability was derived using Online BBB Predictor (https://www.cbligand.org/BBB/, accessed on 30 October 2019), which used 19 simple molecular descriptors for the analysis of 1593 reported compounds, resulting in over 90–95% overall prediction accuracy [29].

### 2.3. Thioflavin T Aggregation Assay

To form Aβ aggregation, Aβ_42_ peptide (10 μM; Kelowna Int’l Scientific Inc., New Taipei City, Taiwan) was incubated with tested compounds (5–20 μM) in 150 mM NaCl and 20 mM Tris-HCl (pH8.0) at 37 °C for 48 h. Thioflavin T (10 μM; Sigma-Aldrich) was added to the Aβ mixture and incubated at room temperature for 5 min. The fluorescence intensity was recorded at 420 nm excitation and 485 nm emission by an FLx800 microplate reader (Bio-Tek, Winooski, VT, USA). Half maximal effective concentration (EC_50_) was estimated by a method of interpolation.

### 2.4. Aβ-GFP Fluorescence and Reactive Oxygen Species (ROS) Analyses

To induce neuronal differentiation, 2 × 10^4^ Aβ-GFP SH-SY5Y cells were seeded on a 96-well plate with retinoic acid (10 µM; Sigma-Aldrich). Cells were pretreated with tested compounds (0.2–5 µM) on day 2 for 8 h and then with doxycycline (5 μg/mL) to induce Aβ-GFP expression. Medium containing retinoic acid compound and doxycycline was refreshed every 2 days for 6 days. Then, cells were stained with Hoechst 33342 (0.1 µg/mL; Sigma-Aldrich) at 37 °C for 30 min. Cell images were recorded at 482/35 nm excitation and 536/40 nm emission wavelengths (ImageXpres Micro Confocal High-Content Imaging System; Molecular Devices, Sunnyvale, CA, USA), and analyzed by MetaXpress (Molecular Devices).

To examine ROS, CellROX Deep Red reagent (5 μM; Molecular Probes, Eugene, OR, USA) was added to the cells and incubated at 37 °C for 30 min. ROS in cells was measured with excitation 631/28 nm and emission 692/40 nm wavelengths.

### 2.5. Neurite Outgrowth Analysis

Aβ-GFP SH-SY5Y cells (6 × 10^4^ cells) were seeded on a 24-well plate with retinoic acid (10 µM) on day 1. Tested coumarins (5 µM) and doxycycline (5 μg/mL) were added on day 2, as described. On day 8, the cells were washed with phosphate-buffered saline (PBS) twice and fixed in 4% paraformaldehyde at 4 °C for 15 min. Cells were permeabilized with Triton X-100 (0.1%) for 10 min, blocked with bovine serum albumin (3%) for 20 min, and stained with anti-neuronal TUBB3 antibody (1:1000; Covance, Princeton, NJ, USA) at 4 °C overnight. The next day, cells were washed with PBS twice and stained by a secondary donkey anti-rabbit Alexa Fluor^®^ 555 antibody (1:1000; Thermo Fisher Scientific) and 4′,6-diamidino-2-phenylindole (DAPI; 0.1 µg/mL; Sigma-Aldrich) at room temperature for 1 h. Neuronal images from at least 60 individual fields (150–250 neurons per field) per experiment were captured at excitation 531/40 nm and emission 593/40 nm wavelengths using an ImageXpress micro confocal high-content imaging system (Molecular Devices). Neurite total length (μm) and numbers of process (the number of primary neurites originated from the cell body of a neuron) and branch (the number of secondary neurites extended from primary neurites) were analyzed using a MetaXpress neurite outgrowth application module (Molecular Devices). For each sample, around 6000 cells were analyzed in each of three independent experiments.

### 2.6. Real-Time PCR Assay

RNA was extracted using TRIzol reagent (Invitrogen, Carlsbad, CA, USA), followed by treatment with DNase I (Stratagene, La Jolla, CA, USA). cDNA was synthesized by SuperScript™ III reverse transcriptase (Invitrogen) according to manufacturer’s instruction and used to determine Aβ-GFP mRNA expression by StepOnePlus™ Real-time PCR system (Applied Biosystems, Foster City, CA, USA) using customized GFP primers (forward primer: 5′-GAGCGCACCATCTTCTTCAAG-3′, reverse primer: 5′-TGTCGCCCTCGAACTTCAC-3′), FAM/NFQ probe (5′-ACGACGGCAACTACA-3′), and TaqMan hypoxanthine phosphoribosyltransferase 1 (HPRT1) endogenous control (VIC/MGB probe, 4326321E) (Applied Biosystems). Fold change was calculated using the formula 2^ΔCt^, ΔC_T_ = C_T_ (HPRT1)—C_T_ (GFP), in which C_T_ indicates the number of threshold cycles.

### 2.7. Caspase-1 and AChE Assays

Cells were lysed using lysis buffer (Caspase-1 Fluorometric Assay Kit; BioVision, Milpitas, CA, USA) over six freeze/thaw cycles. After centrifugation to collect cell lysates, caspase-1 activity in 50 µg cell extracts was measured using YVAD-AFC substrate (AFC: 7-amino-4-trifluoromethyl coumarin). The mixture was incubated at 37 °C for 2 h. AFC fluorescence was measured with 400 nm excitation and 505 nm emission wavelengths (FLx800 fluorescence microplate reader, Bio-Tek).

To measure AChE activity, cells were suspended in cold PBS and lysed by sonication. After centrifugation, the supernatants were collected. AChE activity in 10 µg protein extracts was measured using an AChE activity assay kit (Sigma-Aldrich). The absorbance of the colorimetric product was measured at 412 nm wavelength (Multiskan^TM^ GO spectrophotometer; Thermo Fisher Scientific).

### 2.8. Western Blot Analysis

Cellular proteins were prepared using a lysis buffer containing NaCl (150 mM), ethylene diamine tetraacetic acid (EDTA, 1 mM, pH8.0), Tris-HCl (50 mM, pH8.0), ethylene glycol tetraacetic acid (EGTA, 1 mM, pH8.0), sodium dodecyl sulfate (SDS, 0.1%), sodium deoxycholate (SD, 0.5%), Triton X-100 (1%), and protease (Sigma-Aldrich), and phosphatase (Abcam, Cambridge, MA, USA) inhibitor cocktails. After quantitation by protein assay kit (Bio-Rad, Hercules, CA, USA), 20 µg proteins were separated by SDS-polyacrylamide (10%) gel electrophoresis and transferred to polyvinylidene difluoride (PVDF) membranes (Sigma-Aldrich) using a Mini Trans-Blot cell (Bio-Rad). After blocking, the membrane was probed with anti-TRKB, anti-p-TRKB (Y516), anti-p-TRKB (Y817) (1:500; Cell Signaling, Danvers, MA, USA), anti-ERK, anti-p-ERK (T202/Y204) (1:1000; Cell Signaling), anti-AKT, anti-p-AKT (S473) (1:1000; Cell Signaling), anti-CREB, anti-p-CREB (S133) (1:1000; Millipore, Billerica, MA, USA), anti-BDNF, anti-BCL2 (1:500; Santa Cruz Biotechnology, Santa Cruz, CA, USA), anti-BAX (1:1000; Cell Signaling), anti-β-tubulin (1:500; Sigma-Aldrich), or anti-GAPDH (1:1000, MDBio, Taipei, Taiwan) antibody. The immune complexes were detected using goat anti-mouse or goat anti-rabbit IgG antibody conjugated with horseradish peroxidase (1:5000; GeneTex, Irvine, CA, USA) and chemiluminescent substrate (Millipore).

### 2.9. RNA Interference

Lentiviral short hairpin RNA (shRNA) targeting TRKB (TRCN0000002243, TRCN0000002245, and TRCN0000002246) and a negative-scrambled control (TRC2.Void) were obtained from the National RNAi Core Facility, IMB/GRC, Academia Sinica (Taipei, Taiwan). As described, cells were plated on 6- or 24-well plates, with retinoic acid added on day 1. Cells were infected with lentivirus (multiplicity of infection, 3 for each shRNA), with polybrene (8 µg/mL; Sigma-Aldrich) on the next day. Cells were pretreated with tested compounds (5 µM) for 8 h after changing medium, followed by doxycycline on day 3. Cells were collected for further analysis on day 9. The hairpin sequences of targeting shRNA were below:TRCN0000002243: 5’-CCGGCCAACTATCACATTTCTCGAActcgagTTCGAGAAATGTGATAGTTGGTTTTT-3’TRCN0000002245: 5’-CCGGGCACATCAAGCGACATAACATctcgagATGTT-ATGTCGCTTGATGTGCTTTTT-3’TRCN0000002246: 5’-CCGGCCTTGTTGTATTCCTGCCTTTctcgagAAAGGCAGGAATACAACAAGGTTTTT-3’TRC2.Void: 5’-CCGGAGTTCAGTTACGATATCATGTctcgagACATTCGCGAGTAACTGAACTTTTTT-3’

### 2.10. Parallel Artificial Membrane Permeability Assay (PAMPA)

PAMPA was used to predict the penetration of the tested compounds across the BBB. Briefly, the donor well (Millipore) was filled with 300 μL of the tested compound (1 μM) and QC compounds (carbamazepine, theophylline, or lucifer yellow, 100 μg/mL; Sigma-Aldrich). The filter PVDF membrane (pore size 0.45 μm; Millipore) was coated with 4 μL of porcine polar brain lipid (20 mg/mL; Avanti Polar Lipids, Alabaster, AL, USA) in dodecane and the acceptor well filled with 200 μL of 5% DMSO in PBS. The filter plate was carefully placed on the donor plate to form a sandwich plate at room temperature for 18 h. After the permeation time, the filter and donor plates were separated. The concentration of the tested compound in the donor and acceptor wells was measured by an AB Sciex QTrap 5500 mass spectrometer (Applied Biosystems) linked to a 1200 HPLC system (Agilent Technologies, Palo Alto, CA, USA). The concentrations of the QC compounds were determined by a Tecan Infinite M200 Pro microplate reader (Switzerland). The effective permeability coefficient (P_e_) was calculated as described [30]. Each compound was tested in triplicate.

### 2.11. Statistical Analysis

All experiments were in triplicate. Data are presented as mean ± standard deviation. Differences between groups were evaluated by a two-tailed Student’s *t*-test or one-way analysis of variance with a post hoc Tukey test where appropriate. The level of statistical significance was expressed as a *p*-value less than 0.05.

## 3. Results

### 3.1. Tested Coumarins and Amyloid Inhibition

Five coumarins, ZN009, ZN010, ZN011, ZN014, and ZN015 (Figure 1A), were examined. In a cell culture medium, the five ZN compounds were soluble up to the examined concentration of 100 μM. All five ZN compounds met Lipinski’s rule of 5 guidelines for predicting oral bioavailability (molecular weight ≤450 kDa, hydrogen bond donors ≤5, hydrogen bond acceptors ≤10, calculated octano/water partition coefficient ≤5) [31] (Table 1). In accordance with the calculated polar surface area (PSA) of 63.6–89.9 Å^2^, the five ZN compounds displayed the potential of BBB penetration (<90 Å^2^) [32]. The BBB predictor also suggested BBB-penetration of all examined ZN compounds, with a BBB score 0.04–0.68 times greater than that of the threshold 0.02 [33].

The inhibition of Aβ aggregation was measured by thioflavin T, a dye commonly used to quantitate β-sheet amyloid fibril structures [34]. Curcumin, a well-known inhibitor of Aβ aggregation [35], was included as a positive control. As shown in Figure 1B, Aβ formed aggregates after incubation at 37 °C for 2 days (621 vs. 151 arbitrary units (AUs); *p* < 0.001). Treatment with curcumin at 20 µM decreased the aggregation formation (from 621 to 339 AU; *p* < 0.001). Aβ aggregation was also reduced by ZN015 at 20 µM (from 621 to 403 AU; *p* < 0.001), whereas ZN009, ZN010, ZN011, and ZN014 did not inhibit Aβ aggregation. EC_50_ values of curcumin and ZN015 for Aβ aggregation inhibition were 16 and 30 μM, respectively.

### 3.2. Inhibition Aβ Aggregation and Oxidative Stress by Coumarin Derivatives

SH-SY5Y cells expressing the Aβ-GFP folding reporter [28] were used to evaluate the Aβ aggregation-inhibitory effects of the tested compounds. In these cells, the misfolding and aggregation of Aβ affect the folding of GFP and reduce its fluorescent signal, while the inhibition of Aβ aggregation improves GFP folding and, thereby, increases the fluorescent signal [36]. The Aβ-GFP SH-SY5Y cells were differentiated for 7 days [37], with the induction of Aβ-GFP expression by doxycycline for 6 days (Figure 2A). Under the condition of plating cells and the addition of retinoic acid on day 1 to induce neuronal differentiation, no increased cell density was observed. In addition, neither doxycycline addition nor Aβ induction obviously affected cell viability. As the treatment of curcumin at 10 μM led to appreciable cell death (viability below 80%), 0.2–5 μM concentrations of the compounds, typically in 5-fold dilutions, were selected. After normalization, with cell number counted, treatment with curcumin (111–128%), ZN014 (111–119%), or ZN015 (113–125%) at 1–5 μM significantly increased the GFP fluorescence intensity compared with untreated cells (100%) (*p* = 0.028–0.001) (Figure 2B). No significant change of cell viability was detected (111–96%; *p* > 0.05). Treatment with curcumin, ZN014, or ZN015 at 5 μM did not significantly affect the relative Aβ-GFP/HPRT1 RNA level (29.2–30.0 vs. 28.6 folds of induction) (Figure 2C).

The anti-oxidative effects of curcumin, ZN014, and ZN015 were evaluated by staining the Aβ-GFP SH-SY5Y cells with CellROX Deep Red reagent, a fluorogenic probe commonly used for measurement of intracellular accumulation of radical oxidative species (ROS) [38]. The induction of Aβ-GFP expression increased the ROS level (171%, *p* < 0.001), while treatment with curcumin, ZN014, or ZN015 at 1–5 μM reduced the level of Aβ-induced ROS (from 171% to 146–133%; *p* < 0.001) (Figure 2D).

### 3.3. Inhibition of Caspase-1 and AChE and Promotion of Neurite Outgrowth by Coumarin Derivatives

ROS overproduction induces brain inflammation via the activation of caspase-1, which subsequently induces caspase-6 activation in neurons to lead to axonal degeneration in AD [39]. Inhibition of caspase-1 alleviates neuropathology and improves cognitive function in APP_Sw,Ind_ mice [40]. ROS also induces AChE activity [41], which promotes the assembly of Aβ into oligomers or fibrils [42]. Therefore, the potential of ZN014 and ZN015 to inhibit the activities of caspase-1 and AChE was further evaluated using Aβ-GFP SH-SY5Y cells (Figure 3A,B). The induction of Aβ expression increased caspase-1 (133%, *p* < 0.001) and AChE (123%, *p* = 0.010) activities, while treatment with curcumin, ZN014, and ZN015 (5 µM) rescued the hyperactive caspase-1 (101–95%; *p* < 0.001) and AChE (98–80%; *p* = 0.004–<0.001). 

Aβ aggregation reduces the growth of neurites in primary hippocampal or cortical neurons and Aβ-GFP SH-SY5Y cells [43,44]. As shown in Figure 3C, the overexpression of Aβ reduces the neurite total length (from 31.2 to 23.9 μm, *p* = 0.001), process (from 4.4 to 3.3, *p* = 0.003) and branch (from 2.8 to 1.8, *p* < 0.001). Treatment with curcumin, ZN014, and ZN015 rescued the reduction of neurite length (from 23.9 to 29.1–29.3 μm, *p* = 0.012–0.009), process (from 3.3 to 4.2–4.3, *p* = 0.020–0.007) and branch (from 1.8 to 2.4–2.5, *p* = 0.003–<0.001).

### 3.4. Molecular Targets of New Coumarin Derivatives

BDNF and CREB signaling are involved in neuronal survival, neurite outgrowth and neuroplasticity [13,14]. Therefore, the effects of ZN014 and ZN015 on the expression levels of TRKB, ERK, AKT (Figure 4A), CREB, BDNF, and BCL2 (Figure 4B) were examined. 7,8-DHF, a selective agonist of the TRKB [45], was included for comparison. Treatment with ZN014, ZN015, or 7,8-DHF (5 µM) increased p-TRKB (Y516) (from 40% to 92–100%; *p* = 0.004–0.002), p-TRKB (Y817) (from 71% to 94–108%; *p* = 0.037–0.002), p-AKT (S473) (from 81% to 91–110%; *p* = 0.723–0.023), p-ERK (T202/Y204) (from 79% to 95–107%; *p* = 0.052–0.004), and p-CREB (S133) (from 81% to 98–101%; *p* = 0.028–0.011), as well as downstream BDNF (32 kDa: from 88% to 113–119%, *p* = 0.055–0.016; 14 kDa: from 42% to 124–140%, *p* < 0.001) and BCL2 (from 72% to 104–116%; *p* = 0.015–0.002) protein levels. In response to the anti-apoptotic BCL2 change, treatment with ZN014, ZN015, or 7,8-DHF significantly reduced the expression of pro-apoptotic BAX (from 182% to 126%; *p* = 0.004–0.003).

### 3.5. TRKB Knockdown Attenuated the Neuroprotective Effects of New Coumarin Derivatives

Since the studied ZN014 and ZN015 compounds displayed potential as TRKB agonists, we knocked down TRKB expression to confirm the role of TRKB as the therapeutic target of these compounds (Figure 5A). In scrambled shRNA-infected cells, Aβ overexpression reduced TRKB expression again (80%), but the reduction was not significant (*p* = 0.080). TRKB-specific shRNA further reduced the TRKB level to 32% (*p* < 0.001). Treatment with ZN014, ZN015, or 7,8-DHF did not affect TRKB expression in scrambled shRNA-infected Aβ-expressing cells (77–81%, *p* > 0.05); however, TRKB expression in these cells was counteracted by TRKB-specific shRNA (29–25%, *p* < 0.001) (Figure 5B). Aβ overexpression raised caspase-1 (132%, *p* = 0.005) and AChE (121%, *p* = 0.035) again, but shRNA-mediated TRKB knockdown did not further raise caspase-1 (127%) or AChE (118%) (*p* > 0.05). Treatment with ZN014, ZN015, or 7,8-DHF counteracted the raised caspase-1 (from 132% to 107–105%, *p* = 0.041 –0.019) and AChE (from 121% to 95–91%, *p* = 0.005–0.002) induced by Aβ overexpression. However, TRKB-specific shRNA did not significantly attenuate these rescues (Figure 5C,D).

Neurite outgrowth in Aβ-GFP SH-SY5Y cells is displayed in Figure 5E. Aβ overexpression reduced the neurite length (from 31.2 to 25.3 µm, *p* < 0.001), process (from 4.0 to 3.5, *p* = 0.002), and branch (from 3.0 to 2.4, *p* = 0.002). TRKB-specific shRNA further reduced the neurite length/process/branch to 21.1 µm (*p* = 0.017)/2.9 (*p* = 0.004)/1.9 (*p* = 0.008). Treatment with ZN014, ZN015, or 7,8-DHF rescued the neurite outgrowth in scrambled shRNA-infected cells (length: from 25.3 to 30.7–30.9 µm; process: from 3.5 to 3.9–4.0; branch: from 2.4 to 3.0–3.1; *p* = 0.007–0.001), and in TRKB shRNA-infected cells (length: from 21.1 to 26.9–26.5 µm; process: from 2.9 to 3.5; branch: from 1.9 to 2.6–2.5; *p* = 0.002–<0.001). The improvements of neurite outgrowth by these compounds were partially suppressed by TRKB-specific shRNA (length: from 30.7–30.9 to 26.9–26.5 µm; process: from 3.9–4.0 to 3.5; branch: from 3.0–3.1 to 2.6–2.5; *p* = 0.043–0.010). The results suggested that ZN014, ZN015 and 7,8-DHF exerted neuroprotective effects by upregulating TRKB signaling.

### 3.6. The Potential of BBB Penetration of New Coumarin Derivatives

To confirm the BBB penetration of ZN014, ZN015, and 7,8-DHF, PAMPA [30,46] was employed (Table 2). Quality control compounds, including carbamazepine (a marker for high penetration) [47], theophylline (a marker for low penetration) [48], and lucifer yellow (a marker for integrity) were included. The undetectable penetration of lucifer yellow indicated good membrane integrity. The effective permeability (P_e_) values of carbamazepine and theophylline were 9.85 ± 0.60 and 0.13 ± 0.00 (10^−6^ cm/s), respectively, representing high (>4 × 10^−6^ cm/s) and low (<2 × 10^−6^ cm/s) BBB penetration controls. The P_e_ values of ZN014, ZN015, and 7,8-DHF were 0.56 ± 0.01, 5.16 ± 0.11, and 6.32 ± 1.35 (10^−6^ cm/s), respectively.

## 4. Discussion

Up to the present, effective therapy to slow the progression of neurodegeneration in AD remains an unmet need. Analogs of coumarins showing pharmacological activities have been described [49]. Coumarin and its derivatives demonstrate their potential in treating AD through several mechanisms such as inhibiting AChE [50] and β-secretase [51], preventing misfolded Aβ aggregation [21], upregulating CREB and anti-oxidative pathways [22,23], and promoting BDNF-TRKB and CREB signaling [24,25]. Here, we found the potential of new coumarin derivatives ZN014 and ZN015 for AD treatment by reducing Aβ aggregation, ROS, caspase-1, and AChE as well as promoting neurite outgrowth (Figure 2 and Figure 3) and TRKB signaling (Figure 4). Knockdown of TRKB expression counteracted the neuroprotective effects of these compounds against Aβ toxicity (Figure 5), demonstrating the neuroprotective mechanism of ZN014 and ZN015 is mediated by enhancing TRKB signaling. It is noted that the knockdown of TRKB did not increase the activity of caspase-1 and AChE. These may be explained by the fact that caspase-1 and AChE activity are elevated mainly by other mechanisms such as increased oxidative stress and inflammation and not by decreased TRKB in the SH-SY5Y cells expressing Aβ-GFP. Our study results are supported by a previous study that has also shown AChE activity is not affected by deficient TRKB [52]. Moreover, the partial neurite outgrowth rescue effects of ZN014 and ZA015 in cells with knockdown of TRKB also indicate the contribution of other signaling pathways to the neuroprotection of these compounds.

Oxidative stress has been identified as an important factor contributing to the neurodegeneration of AD [53]. Compounds with anti-oxidative potential may directly serve as chemical chaperones to suppress protein aggregates, quench free oxygen radicals, or enhance anti-oxidative signaling to influence cellular ROS [54,55]. In our study, only ZN015 displayed chemical chaperone activity for Aβ aggregation (Figure 1B), and both ZN014 and ZN015 showed no 1,1-diphenyl-2-picrylhydrazyl radical scavenging activity against ROS (data not shown). As coumarin and its derivatives demonstrate the potential to activate NRF2 anti-oxidative signaling in different cells and animal models [22,56], the anti-oxidative effect of ZN014 and ZN015 in our cell model (Figure 2E) may also be upregulated by anti-oxidative signaling.

The production of ROS by Aβ aggregation upregulates caspase-1 activity and induces neuroinflammation [57]. Caspase-1 is involved in the cleavage and activation of interleukins 1β, 18, and 33 [58]. In addition, caspase-1 induces caspase-6 activation, leading to axonal degeneration [39], and axonopathy is recognized as an early event of patients with AD [59]. Axonal degeneration, with swellings of haphazardly arranged vesicles, mitochondria, multilamellar bodies, and vacuoles, and impaired axonal transport could be observed to precede the development of amyloid plaques in the Tg-swAPP^Prp^ mouse model for AD [59]. Activation of caspase-1 also induces pyroptosis with the secretion of TNF-α and IL-6 [60]. Inhibition of caspase-1 reverses memory impairment and decreases Aβ accumulations and neuroinflammation in the brains of the caspase-1 null J20 mouse model of AD [40]. The coumarin derivative nodakenin has been reported to inhibit the production of cytokines via the suppression of caspase-1 activation in anaphylactic mice [61]. In our study, both ZN014 and ZN015 counteracted the Aβ-induced increase in caspase-1 activity (Figure 3A).

AChE, an enzyme breaking down acetylcholine into acetate and choline, also accelerates the formation of Aβ fibrils [62]. ACE inhibitors may improve AD neurodegeneration by increasing the level and action duration of acetylcholine [63] as well as reducing the formation of Aβ aggregation [64]. AChE-inhibitory activities of coumarin derivatives have been reported [50]. Resembling coumarin and LM-031 [23], ZN015 exhibited inhibitory activity on both AChE (Figure 3B) and Aβ aggregation (Figure 1B).

Upon BDNF binding, TRKB dimerizes and phosphorylates to initiate intracellular ERK and AKT signaling, leading to CREB phosphorylation for the survival of neurons [65]. Upon the phosphorylation of serine at position 133 (S133), phosphor-CREB translocates to the nucleus and binds to a cAMP-response-element (CRE) [66], thereby inducing the expression of CRE-mediated transcription of genes such as neurotrophic BDNF and BCL2 for neuroprotection [67]. BCL2 prevents BAX redistribution to the mitochondria, where it forms oligomers, resulting in the efflux of cytochrome c and the induction of the apoptotic cascade [68]. In human neurons, Aβ downregulates BCL2 and increases the level of BAX [69]. In Aβ-GFP SH-SY5Y cells, induction of Aβ-GFP expression downregulated BCL2 and upregulated BAX, and ZN014 and ZN015 counteracted changes in gene expression for these CREB-responsive genes (Figure 4). Of note, ZN014 and ZN015 also upregulated the expression of BDNF (Figure 4), forming positive feedback in the BDNF–TRKB–CREB signaling pathway.

Finally, it is well noted that pre-conditioning cellular protection through NRF2 anti-oxidative signaling has the hormesis feature [70]. Hormesis is an adaptive biological response to drugs or treatment, which indicates that a greater magnitude of therapeutic effect was seen at the middle dose range and a less protective effect, with stronger cell toxicity, was seen at the higher doses of a compound (a specific pattern of biphasic dose–response of a compound) [71]. The hormesis of anti-oxidative gene networks in redox reactions is also important for dose optimization in treating neurodegenerative diseases [72]. Further study will be needed to explore the interplay between antioxidant signaling and other signals by coumarin derivatives.

## 5. Conclusions

In conclusion, we found the neuroprotective potential of two new coumarin derivatives, ZN014 and ZN015, against Aβ neurotoxicity via the inhibition of oxidative stress, caspase-1, and AChE activities and the activation of TRKB signaling in the Aβ-GFP SH-SY5Y cell model (Figure 6). As AD has complex neurodegenerative pathogenesis, the pleiotropic mechanism of ZN014 and ZN015 make these compounds promising for drug development. However, the SH-SY5Y cell model only emphasizes the degeneration of neurons, while the pathogenesis of AD also involves glial cells such as astrocytes and microglia. The interactions between neurons and glial cells are also not addressed in this cell model. Although ZN014 and ZN015 rescued the neurite outgrowth deficit after Aβ induction, we did not show if those compounds had a neurotrophic effect on neurite outgrowth without Aβ induction. Given that in clinical practice, we will not treat healthy individuals with drugs, we consider that the effects of the compounds on neurite outgrowth in cells without Aβ induction may not be crucial and the experiment could have been skipped in this study. Furthermore, our findings are limited in human cell models. Future validation in AD animal models will be conducted. The binding of ZN014 and ZN015 to TRKB will also be measured using surface plasmon resonance to consolidate their properties as TRKB agonists.

## Figures and Tables

**Figure 1 cells-10-03095-f001:**
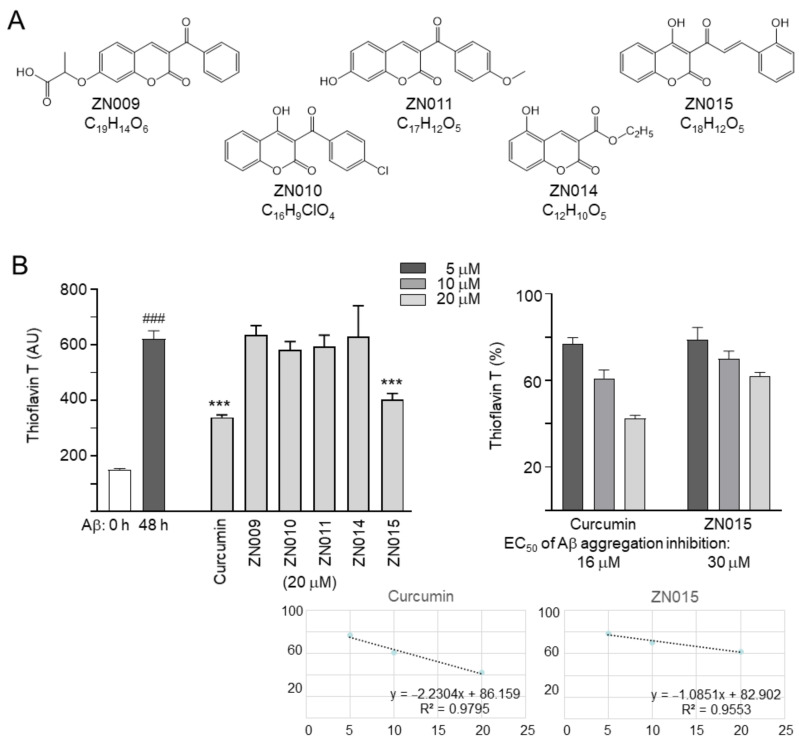
Tested ZN compounds and Aβ aggregation inhibition. (**A**) Structure and formula of ZN009, ZN010, ZN011, ZN014, and ZN015. (**B**) Left: Aβ aggregation inhibition of curcumin (known Aβ aggregation inhibitor) and ZN compounds (20 μM) by thioflavin T fluorescence assay (*n* = 3). *p*-values: comparisons between with and without 2 days’ incubation at 37 °C (^###^: *p* < 0.001) or with and without compound addition (***: *p* < 0.001; one-way ANOVA with a post hoc Tukey test). Right: Thioflavin T assay for Aβ aggregation inhibition by curcumin and ZN015 (5–20 μM) (*n* = 3), with EC_50_ values estimated with the interpolation of the straight line below. The relative thioflavin T fluorescence without compound addition was normalized as 100%.

**Figure 2 cells-10-03095-f002:**
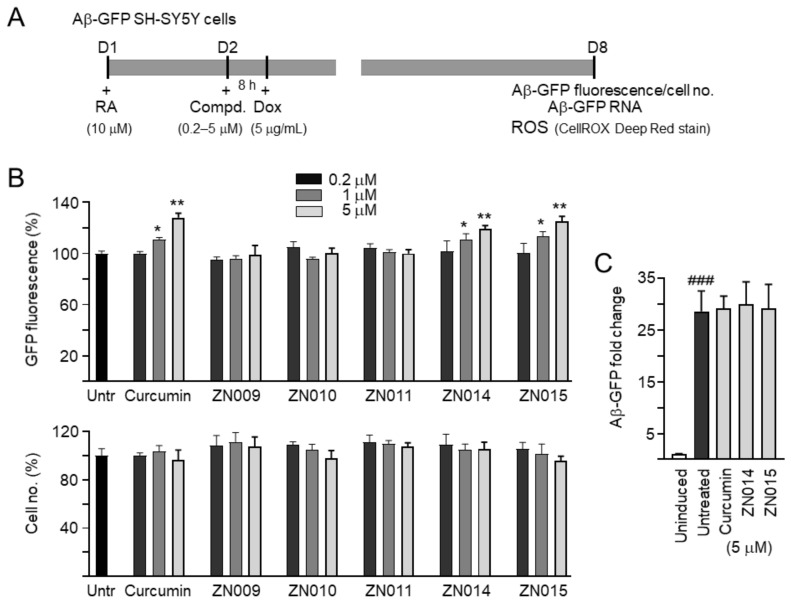
Aβ aggregation inhibition and ROS reduction of tested coumarins in SH-SY5Y cells expressing Aβ-GFP. (**A**) Experimental flow chart. Cells were plated with retinoic acid (RA, 10 µM) addition on day 1 and treated with tested compounds (0.2–5 µM) for 8 h, followed by inducing Aβ-GFP expression with doxycycline (Dox, 5 µg/mL) on day 2. Aβ-GFP fluorescence, cell number, induced Aβ-GFP RNA, and ROS were measured on day 8. (**B**) Quantitation of GFP fluorescence and cell number in Aβ-GFP–expressing cells untreated or treated with curcumin, ZN009, ZN010, ZN011, ZN014, or ZN015 at 0.2–5 µM (*n* = 3). The relative GFP fluorescence and cell number of untreated cells were normalized as 100%. *p-*values: comparisons between with and without compound addition (*: *p* < 0.05, **: *p* < 0.01; two-tailed Student’s *t*-test). Shown on the bottom were GFP (green) images of Aβ-GFP–expressing cells untreated or treated with 5 µM curcumin, ZN014, and ZN015. Nuclei were counterstained with Hoechst 33342 (blue). (**C**) Real-time PCR analysis of Aβ-GFP RNA in SH-SY5Y cells uninduced, untreated, or treated with 5 µM curcumin, ZN014, or ZN015 (*n* = 3). HPRT1 was used as an endogenous control to normalize between samples. (**D**) Quantitation of ROS (*n* = 3), with relative ROS in uninduced cells normalized (100%). Shown at the bottom are images (CellROX Deep Red, red) of SH-SY5Y cells uninduced, untreated, or treated with the 5 µM compound. Nuclei were counterstained with Hoechst 33342 (blue). (**C**,**D**) *p*-values: comparisons between untreated vs. uninduced cells (^###^: *p* < 0.001) or compound-treated vs. untreated cells (***: *p* < 0.001) (one-way ANOVA with a post hoc Tukey test).

**Figure 3 cells-10-03095-f003:**
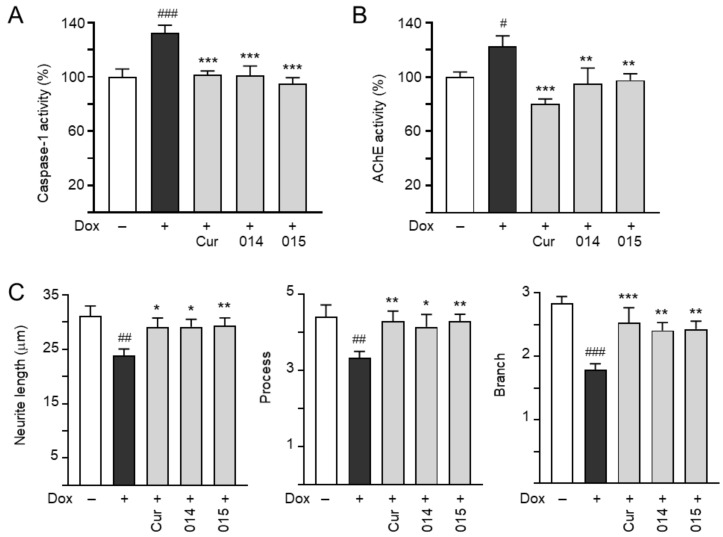
Caspase-1, AChE reduction, and neurite outgrowth promotion of ZN014 or ZN015 in SH-SY5Y cells expressing Aβ-GFP. As described, cells were seeded with retinoic acid, treated with curcumin, ZN014, or ZN015 (5 µM), and induced for Aβ-GFP expression with doxycycline for 6 days. On day 8, (**A**) caspase-1 activity, (**B**) AChE activity, and (**C**) neurite length, process, and branch were analyzed (*n* = 3). The relative caspase-1 or AChE activity of uninduced cells (Dox −) was normalized (100%). Shown at the bottom of (**C**) are images of class III β-tubulin (TUBB3) (yellow)-stained cells, with nuclei counterstained with DAPI (blue), and segmented images with a multicolored mask to assign each outgrowth to a cell body for neurite outgrowth quantification. In uninduced cells, processes and branches are indicated with red and white arrows, respectively. *p*-values: comparisons between induced vs. uninduced cells (^#^: *p* < 0.05, ^##^: *p* < 0.01, ^###^: *p* < 0.001) or compound-treated vs. untreated (induced) cells (*: *p* < 0.05, **: *p* < 0.01, ***: *p* < 0.001) (one-way ANOVA with a post hoc Tukey test).

**Figure 4 cells-10-03095-f004:**
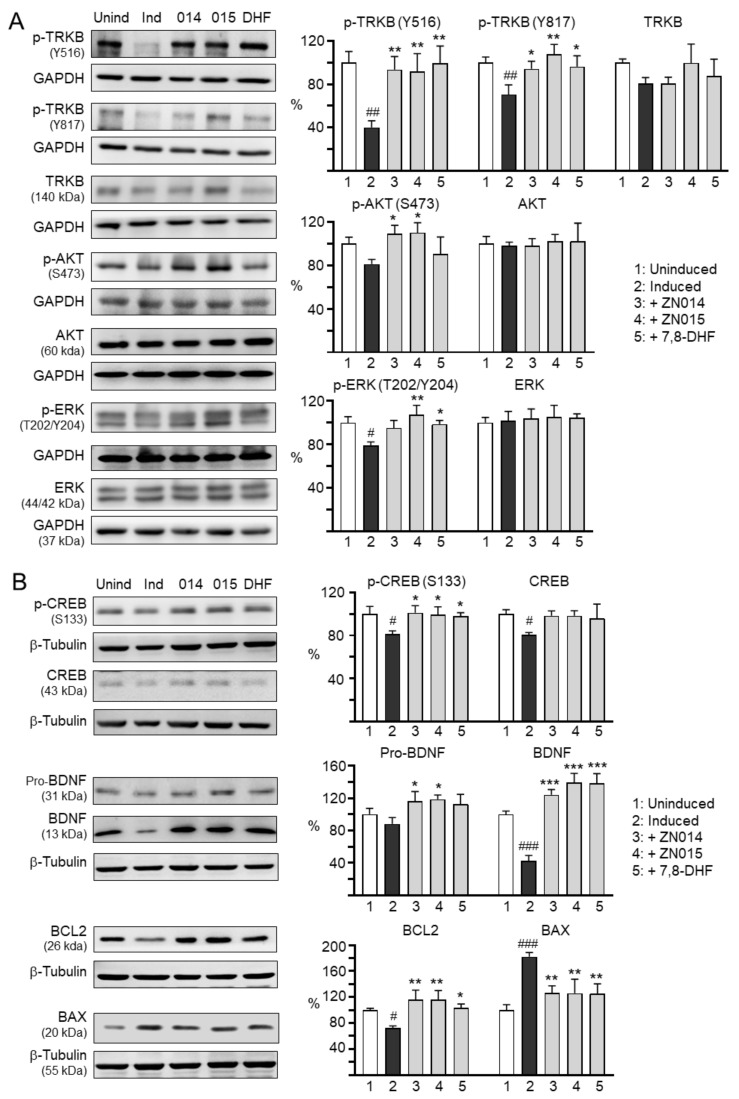
Molecular targets of ZN014 and ZN015 in SH-SY5Y cells expressing Aβ-GFP. 7,8-DHF (5 µM) was included as a positive control. Relative (**A**) TRKB, p-TRKB (Y516 and Y817), AKT, p-AKT (S473), ERK, p-ERK (T202/Y204), (**B**) CREB, p-CREB (S133), precursor pro-BDNF, mature BDNF, BCL2, and BAX protein levels were analyzed by immunoblotting using specific antibodies (*n* = 3). Glyceraldehyde-3-phosphate dehydrogenase (GAPDH) (**A**) or β-tubulin (**B**) was used as a loading control. Relative protein levels are shown on the right side of the representative Western blot images. The relative protein level in uninduced cells (Dox −) was normalized (100%). *p*-values: comparisons between induced and uninduced cells (^#^: *p* < 0.05, ^##^: *p* < 0.01, ^###^: *p* < 0.001), or between treated and untreated cells (*: *p* < 0.05, **: *p* < 0.01, ***: *p* < 0.001) (one-way ANOVA with a post hoc Tukey test).

**Figure 5 cells-10-03095-f005:**
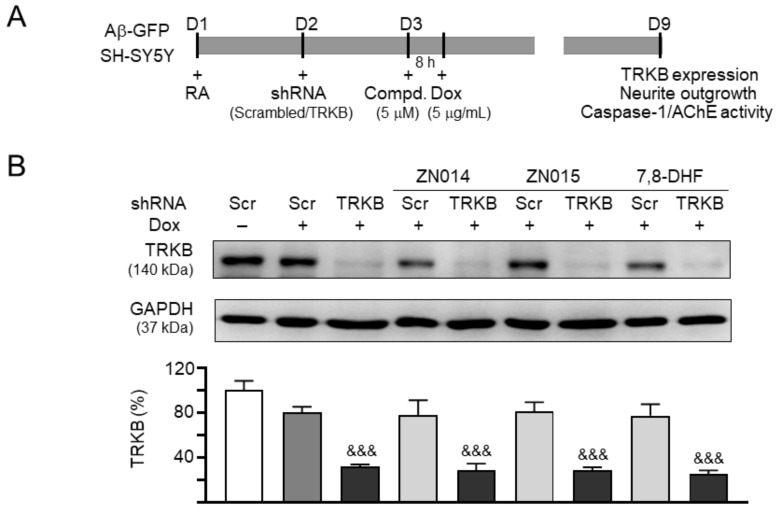
TRKB RNA interference of SH-SY5Y cells expressing Aβ-GFP. (**A**) Experimental flow chart. On day 1, Aβ-GFP SH-SY5Y cells were plated with retinoic acid (RA; 10 µM). On day 2, cells were infected by lentivirus-expressing TRKB-specific or scrambled shRNA. One day post-infection, 5 µM ZN014, ZN015, or 7,8-DHF (as a positive control) was added to the cells for 8 h, followed by the induction of Aβ-GFP expression (Dox, 5 µg/mL) for 6 days. On day 9, the cells were collected for (**B**) TRKB protein (GAPDH as a loading control), (**C**) caspase-1 activity, (**D**) AChE activity, and (**E**) neurite outgrowth analyses (*n* = 3). The relative TRKB protein, caspase-1, or AChE activity of uninduced cells (Dox −) was normalized (100%). Shown on the bottom of (**E**) were images of TUBB3 (yellow)-stained cells, with nuclei counterstained with DAPI (blue), and segmented images with a multicolored mask to assign each outgrowth to a cell body for neurite outgrowth quantification. Processes and branches in scrambled shRNA-infected uninduced cells are marked with red and white arrows, respectively. *p*-values: comparisons between induced (Dox +) vs. uninduced (Dox −) cells (^#^: *p* < 0.05, ^##^: *p* < 0.01, ^###^: *p* < 0.001), between compound-treated vs. untreated cells infected with scrambled shRNA (*: *p* < 0.05, **: *p* < 0.01) or between compound-treated vs. untreated cells infected with TRKB shRNA (^^: *p* < 0.01, ^^^: *p* < 0.001) or between TRKB shRNA-treated vs. scrambled shRNA-treated cells (^&^: *p* < 0.05, ^&&^: *p* < 0.01, ^&&&^: *p* < 0.001) (one-way ANOVA with a post hoc Tukey test).

**Figure 6 cells-10-03095-f006:**
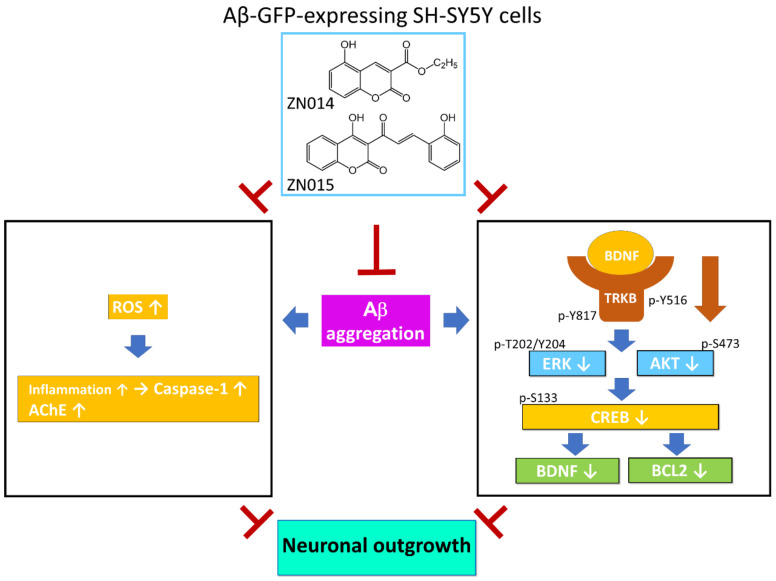
Model for Aβ aggregation reduction and neuronal outgrowth promotion by synthetic coumarin derivatives ZN014 and ZN015 in an Aβ-GFP–expressing SH-SY5Y AD cell model. ZN014 and ZN015 inhibit ROS to reduce caspase-1 and AChE activities in SH-SY5Y cells, both of which would otherwise worsen Aβ-induced pathology. In addition, ZN014 and ZN015 activate TRKB-CREB signaling in SH-SY5Y cells to promote neuronal outgrowth.

**Table 1 cells-10-03095-t001:** Bioavailability and BBB permeation prediction of ZN compounds.

Compound Name	MW	HBD	HBA	cLogP	PSA (Å^2^)	BBB Score (Threshold: 0.02)
ZN009	338.31	1	6	2.44	89.9	0.68
ZN010	300.69	1	4	4.23	63.6	0.15
ZN011	296.28	1	5	2.74	72.8	0.04
ZN014	234.21	1	5	2.17	72.8	0.19
ZN015	308.29	2	5	3.28	83.8	0.14

MW, Molecular weight; HBD, hydrogen bond donor; HBA, hydrogen bond acceptor; cLogP, calculated octanol-water partition coefficient; PSA, polar surface area; BBB, blood–brain barrier.

**Table 2 cells-10-03095-t002:** Permeability of ZN014, ZN015, 7,8-DHF and the QC compound by the PAMPA-BBB method.

Compound Name	Measured P_e_ (10^−6^ cm/s) or % Transport	BBB Permeability Classification *^a^*
ZN014	0.56 ± 0.01	Low
ZN015	5.16 ± 0.11	High
7,8-DHF	6.32 ± 1.35	High
Carbamazepine	9.85 ± 0.60	High marker
Theophylline	0.13 ± 0.00	Low marker
Lucifer yellow	0.00 (% Transport)	Integrity marker (well-accepted membrane integrity)

*^a^* Suggested BBB permeability: P_e_ > 4 × 10^−6^ cm/s (high), 4 > P_e_ > 2 × 10^−6^ cm/s (moderate), and P_e_ < 2 × 10^−6^ cm/s (low).

## Data Availability

The data generated during the study are available from the corresponding author upon request.

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
