# Peer review of "Multi-Target Effects of Novel Synthetic Coumarin Derivatives Protecting Aβ-GFP SH-SY5Y Cells against Aβ Toxicity"

_cells, 2021, doi:10.3390/cells10113095_

Round 1

Reviewer 1 Report

In numerous experimental models, natural antioxidants are shown  to induce hormetic dose responses that are not only common but display endpoints of biomedical and clinical relevance. These hormetic responses are mediated via the activation of nuclear factor erythroid- derived 2 (Nrf2) antioxidant response elements (AREs) and, as such, are characteristically biphasic, well integrated, concentration/dose dependent, and specific with regard to the targeted cell type and the temporal profile of response. In experimental disease models, the polyphenol-induced hormetic activation of Nrf2 was shown to effectively reduce the occurrence and severity of a wide range of human-related pathologies, including Parkinson's disease, Alzheimer's disease, stroke, age-related ocular damage, chemically induced brain damage, and renal nephropathy, amongst others, while also enhancing stem cell proliferation. Interestingly, the mechanistic profile is similar to that of numerous other hormetic agents, indicating that activation of the Nrf2/ARE pathway is probably a central, integrative, and underlying mechanism of hormesis itself. The Nrf2/ARE pathway provides an explanation for how large numbers of agents that both display hormetic dose responses and activate Nrf2 can function to limit age-related damage, the progression of numerous disease processes, and chemical- and radiation- induced toxicities. These findings extend the generality of the hormetic dose response to include many other chemical activators of Nrf2 that are cited in the biomedical literature and therefore have potentially important public health and clinical implication. Thus, Interplay and coordination of redox interactions with endogenous and exogenous antioxidant defence systems  is an emerging area of reserach interest in anticancer and antidegenerative therapeutics. Moreover, particular attention has been given to providing an assessment of the quantitative features of the dose-response relationships and underlying mechanisms that could account for the biphasic nature of the hormetic response after exposure to redox active agents, such as free radical oxygen species and their impact in inflammatory/antinflammatory pathways. The hormetic dose response should be seen as a reliable feature of the dose response for oxygen free radicals and their redox regulated transcriptional factors  as well as  antioxidant compounds and appears to have an important impact  on brain pathophysiology and stress resistance mechanisms to oxidative and inflammatory insult and neurodegenerative damage.
This is an interesting paper.  The study is well-conceived and well-executed. This reviewer is satisfied with the significance of this study, the care in which the study was performed, and the implications of the results for human health.  However, although the results presented are convincing, the work raises some concerns which will need to be addressed. The questions posed are of extremely high interest, but the paper does not give adequate definitive information, therefore pending addressing some major question is possible to accept for publication.

Minor concerns:

1.    Given the relationship between polyphenol compounds, redox status and the vitagene network and its possible biological relevance in neuroprotection, Authors while interpetrating results should discuss appropriately this aspect and make proper connection with emerging principles of hormesis. Indeed, preconditioning signal leading to cellular protection through Hormesis is an important redox dependent aging-associated to free radicals species accumulation, inflammatory responses involved in neurodegenerative/ neuroprotective mechanisms. This aspect should be highlighted in the discussion and references properly added (See Calabrese et al., 2010, Antiox. Redox Signal 13,1763; Calabrese et al., 2000 J. Neurosci. Res. 60, 613; Dattilo et al., 2015, Immun  Ageing. 12:20. doi: 10.1186/s12979-015-0046-8.

Author Response

We added a paragraph in lines 449-456 to address this: Finally, it is well noted that pre-conditioning cellular protection through NRF2 anti-oxidative signaling has the hormesis feature [70]. Hormesis is an adaptive biological response to drugs or treatment, which indicates that a greater magnitude of therapeutic effect was seen at the middle dose range and a less protective effect with stronger cell toxicity was seen at the higher doses of a compound (a specific pattern of biphasic dose response of a compound) [71]. The hormesis of anti-oxidative gene networks in redox reactions are also important for dose optimization in treating neurodegenerative diseases [72]. Further study will be demanded to explore the interplay between antioxidant signaling and other signals by coumarin derivatives.

[70] Siracusa, R.; Scuto, M.; Fusco, R.; Trovato, A.; Ontario, M.L.; Crea, R.; Di Paola, R.; Cuzzocrea, S.; Calabrese, V. Anti-inflammatory and anti-oxidant activity of Hidrox® in rotenone-induced Parkinson's disease in mice. Antioxidants 2020, 9, 824. [CrossRef] [PubMed]

[71] Miquel, S.; Champ, C.; Day, J.; Aarts, E.; Bahr, B.A.; Bakker, M.; Bánáti, D.; Calabrese, V.; Cederholm, T.; Cryan, J.; et al. Poor cognitive ageing: Vulnerabilities, mechanisms and the impact of nutritional interventions. Ageing. Res. Rev. 2018, 42, 40-55. [CrossRef] [PubMed]

[72] Brunetti, G.; Di Rosa, G.; Scuto, M.; Leri, M.; Stefani, M.; Schmitz-Linneweber, C.; Calabrese, V.; Saul, N. Healthspan maintenance and prevention of Parkinson's-like phenotypes with hydroxytyrosol and oleuropein aglycone in C. elegans. Int. J. Mol. Sci. 2020, 21, 2588. [CrossRef] [PubMed]

Reviewer 2 Report

Huang et al showed two synthesized compounds were against Aβ toxicity including increasing caspase-1 and AChE and inhibiting neurite outgrowth in SH-SY5Y cells and BDNF-TRKB signaling mediated the protective effects on neurite outgrowth. This is a well-design study and the conclusions are sound. Some concerns need to be addressed.

  1. Experimental flow chart is needed for figure 1C. Whether doxycycline was used in these experiments in figure1C?
  2. Line180: what are the sequences of shRNA ?
  3. Line225: how to calculate the EC50 values? dose–response curve needs to be shown.
  4. EC50 values of curcumin and ZN015 for Aβ aggregation inhibition were 16 μM and 30 μM, respectively. Why 1–5 μM was used in the following experiments ? how to determine the selected concentration for the compounds?
  5. Figure2C: Y axis label should be fold change.
  6. Line401: remove ” AD”
  7. SH-SY5Y is a proliferating cell line. The authors cultured for 8-9 days without splitting before performing experiments. Whether the cell density increased ?
  8. In Fig2B, doxycycline did not change the cell number. Whether Aβ induction changed the cell viability?
  9. Fig3B: ZN014 and ZN015 rescued neurite outgrowth deficit after Aβ induction. Whether those compounds had neurotrophic effect on neurite outgrowth without Aβ induction ?
  10. Neurite outgrowth in AD pathology should be discussed.
  11. The limitation of using SH-SY5Y as AD model should be illustrated.
  12. No pro-survival evidence provided in the study. The model in figure 6 need to be modified.

Author Response

Concerns addressed:

1. Experimental flow chart is needed for figure 1C. Whether doxycycline was used in these experiments in figure1C?

The thioflavin T (ThT) assay in figure 1C measures changes of fluorescence intensity of ThT upon binding to amyloid fibrils in the presence or absence of tested compounds. No cells or doxycycline were involved and no experimental flow chart is needed.

2. Line180: what are the sequences of shRNA ?

We added the sequences of shRNA in lines 179-188: The hairpin sequences of targeting shRNA were below

TRCN0000002243: 5'-CCGGCCAACTATCACATTTCTCGAActcgagTTCGAGAAATGTGATAGTTGGTTTTT-3'

TRCN0000002245: 5'-CCGGGCACATCAAGCGACATAACATctcgagATGTTATGTCGCTTGATGTGCTTTTT-3'

TRCN0000002246: 5'-CCGGCCTTGTTGTATTCCTGCCTTTctcgagAAAGGCAGGAATACAACAAGGTTTTT-3'

TRC2.Void: 5'-CCGGAGTTCAGTTACGATATCATGTctcgagACATTCGCGAGTAACTGAACTTTTTT-3'

3. Line225: how to calculate the EC50 values? dose–response curve needs to be shown.

We showed dose–response curves in Figure 1C.

4. EC50 values of curcumin and ZN015 for Aβ aggregation inhibition were 16 μM and 30 μM, respectively. Why 1–5 μM was used in the following experiments ? how to determine the selected concentration for the compounds?

EC50 values of 16-30 μM for curcumin and ZN015 were biochemical measurements of thioflavin T binding to amyloid fibrils. For cellular assay, we added a sentence in lines 246-248 to explain the selected concentration for the compounds: As treatment of curcumin at 10 μM led to appreciable cell death (viability below 80%), 0.2–5 μM concentrations of compounds, typically in 5-fold dilutions, were selected.

5. Figure2C: Y axis label should be fold change.

We changed Figure 2C Y axis label Aβ-GFP/HPRT1 to Aβ-GFP fold change.

6. Line401: remove ” AD”

We remove (DPPH) in line 415.

7. SH-SY5Y is a proliferating cell line. The authors cultured for 8-9 days without splitting before performing experiments. Whether the cell density increased ?

We added a sentence in lines 244-245 to address this: Under the condition of plating cells and addition of retinoic acid on day 1 to induce neuronal differentiation, no increased cell density was observed.

8. In Fig2B, doxycycline did not change the cell number. Whether Aβ induction changed the cell viability?

We added a sentence in lines 245-246 to address this: In addition, neither doxycycline addition nor Aβ induction affected obviously the cell viability.

9. Fig3B: ZN014 and ZN015 rescued neurite outgrowth deficit after Aβ induction. Whether those compounds had neurotrophic effect on neurite outgrowth without Aβ induction ?

We added sentences in lines 465-469 to address this: Although ZN014 and ZN015 rescued neurite outgrowth deficit after Aβ induction, we did not show if those compounds had neurotrophic effect on neurite outgrowth without Aβ induction. Given that in clinical practice, we won’t treat healthy individuals with drugs, we consider the effects of the compounds on neurite outgrowth in cells without Aβ induction may not be crucial and the experiment may be skipped in this study.

10. Neurite outgrowth in AD pathology should be discussed.

We added sentences in lines 421-425 to address this: In addition, caspase-1 induces caspase-6 activation leading to axonal degeneration [39], and axonopathy was recognized as an early event of patients with AD [59]. Axonal degeneration with swellings of haphazardly arranged vesicles, mitochondria, multilamellar bodies and vacuoles, and impaired axonal transport could be observed preceding the development of amyloid plaques in Tg-swAPPPrp mouse model for AD [59].

11. The limitation of using SH-SY5Y as AD model should be illustrated.

We added sentences in lines 462-465 to address this: However, SH-SY5Y cell model only emphasizes the degeneration of neurons, while the pathogenesis of AD is also involving glial cells such as astrocytes and microglia. The interactions between neurons and glial cells are also not addressed in this cell model.

12. No pro-survival evidence provided in the study. The model in figure 6 need to be modified.

We remove survival in Figure 6 and legend.